# Multi-Organ and Pan-cancer Segmentation Framework from Partially Labeled Abdominal CT Datasets: Fine and Swift nnU-Nets with Label Fusion

Youngbin Kong[1,2][0009−0002−6248−4269], Kwangtai Kim[2][0009−0009−1141−5985], Seoi Jeong[2][0009−0009−3047−2889], Kyu Eun Lee[3,4][0000−0002−2354−3599], and Hyoun-Joong Kong[2,3,5,⋆][0000−0001−5456−4862]

[1] Interdisciplinary Program in Bioengineering, Graduate School, Seoul National University, Seoul, Republic of Korea
[2] Department of Transdisciplinary Medicine, Seoul National University Hospital, Seoul, Republic of Korea
[3] Medical Big Data Research Center, Seoul National University College of Medicine, Seoul, Republic of Korea
[4] Department of Surgery, Seoul National University Hospital and College of Medicine, Seoul, Republic of Korea
[5] Department of Medicine, Seoul National University College of Medicine, Seoul, Republic of Korea
gongcop7@snu.ac.kr

**Abstract.** Segmentation of organs and tumors from abdominal computed tomography (CT) scans is crucial for cancer diagnosis and surgical planning. Since traditional segmentation methods are subjective and labor-intensive, deep learning-based approaches have been introduced recently which incur high computational costs. This study proposes an accurate and efficient segmentation method for abdominal organs and tumors in CT images utilizing a partially-labeled abdominal CT dataset. Fine nnU-Net was used for the pseudo-labeling of unlabeled images. And the Label Fusion algorithm combined the benefits of the provided datasets to build an optimal training dataset, using Swift nnU-Net for efficient inference. In online validation using Swift nnU-Net, the dice similarity coefficient (DSC) values for organs and tumors segmentation were 89.56% and 35.70%, respectively, and the normalized surface distance (NSD) values were 94.67% and 25.52%. In our own efficiency experiments, the inference time was an average of 10.7 seconds and the area under the GPU memory time curve was an average of 20316.72MB. Our method enables accurate and efficient segmentation of abdominal organs and tumors using partially labeled data, unlabeled data, and pseudo-labels. This method could be applied to multi-organ and pan-cancer segmentation in abdominal CT images under low-resource environments.

**Keywords:** Label Fusion · Abdominal CT · Segmentation · Partially Labeled Dataset · Deep Learning

---

⋆ corresponding author

## 1   Introduction

In the medical field, the precise segmentation of organs and tumors in abdominal images from medical imaging modalities, such as computed tomography (CT), magnetic resonance imaging (MRI), constitutes a pivotal and indispensable undertaking. This crucial process plays a pivotal role in the diagnosis and management of cancer, encompassing both treatment planning and execution, as well as ongoing patient monitoring [8]. Patient-specific anatomical models based on segmentation are used in the surgical planning phase and during surgical procedures. Especially CT should accurately segment multiple organs and tumors in the abdominal region within a CT image, owing to its critical use in many medical diagnoses. However, due to low-contrast binary CT images [24], traditional manual segmentation can be subjective when outlining soft tissues, such as organs [5], resulting in inconsistent results and significant labor and expertise. Based on these limitations, recent research trends have focused on deep-learning-based methods, such as nnU-Net [15], UNETR [10], EfficientSeg [25], V-Net [19], and Med3D [2], to segment multiple organs and tumors in the abdominal region of CT images. Furthermore, to intricately segment the complex structures of the abdomen and tumors, models based on convolutional neural networks are equipped with sophisticated architectures designed. However, these findings are limited to specific organs and their associated tumors, including liver and kidney tumors [9,12]. Comprehensive studies addressing the segmentation of multiple organs and tumors throughout the abdomen are limited. Furthermore, producing fully labeled datasets still relies on traditional annotation techniques, focusing on expensive supervised and semi-supervised learning [16]. Consequently, studies based on partial labels, in which only some images are annotated, are becoming increasingly important [27].

Most deep-learning-based medical image analysis tasks focusing on high-resolution, large-capacity three-dimensional image data and high-performance models require considerable computational time and graphical processing unit (GPU) resources [3]. However, owing to the possibility of an urgent surgery, hospitals should promptly provide accurate segmentation results.

Thus, FLARE22 focused on a semi-supervised segmentation task that required a fully labeled dataset for multiple organs, whereas FLARE23 extended the topic to a partial-label segmentation task for multiple organs and tumors. Additionally, it provides partially labeled and unlabeled images. Moreover, pseudo-labels generated by models from FLARE22, which had demonstrated superior accuracy for the entire image set, are also being offered.

In this study, we propose a method to perform fast and accurate segmentation of abdominal organs and tumors based on the nnU-Net, which has attracted attention for overall medical image segmentation problems. Compared to conventional U-Net, nnU-Net is a model with wider scalability in medical image analysis, which has an encoder-decoder structure similar to U-Net and applies techniques such as skip connections. Variables in this nnU-Net that impact accuracy and efficiency were identified and adjusted to construct our model. Our methodology consists of a 'Fine nnU-Net' designed to make precise predictions

for high-quality pseudo-labeling of unlabeled images, a 'Label Fusion Algorithm' that combines different types of labels to create meaningful labels, and a "Swift nnU-Net" that is lightly optimized for fast inference. For efficient prediction, we adopted methods proposed in FLARE22 such as the efficient sliding window technique [14].

Our contributions are as follows:

- **Utilization and Advancement of nnU-Net:** We propose an enhanced-accuracy segmentation method for abdominal organs and tumors based on the pivotal nnU-Net in the medical image segmentation field.
- **Effective Label Processing Methodology:** To effectively combine various labels, we propose Fine nnU-Net for high-quality pseudo-labeling for unlabeled images by leveraging partially annotated images and Label Fusion Algorithm.
- **Optimization in Low-Resource Environment:** Using the optimized "Swift nnU-Net," we enable fast inference and suggest model optimizations to function efficiently even in limited computational resource environment.

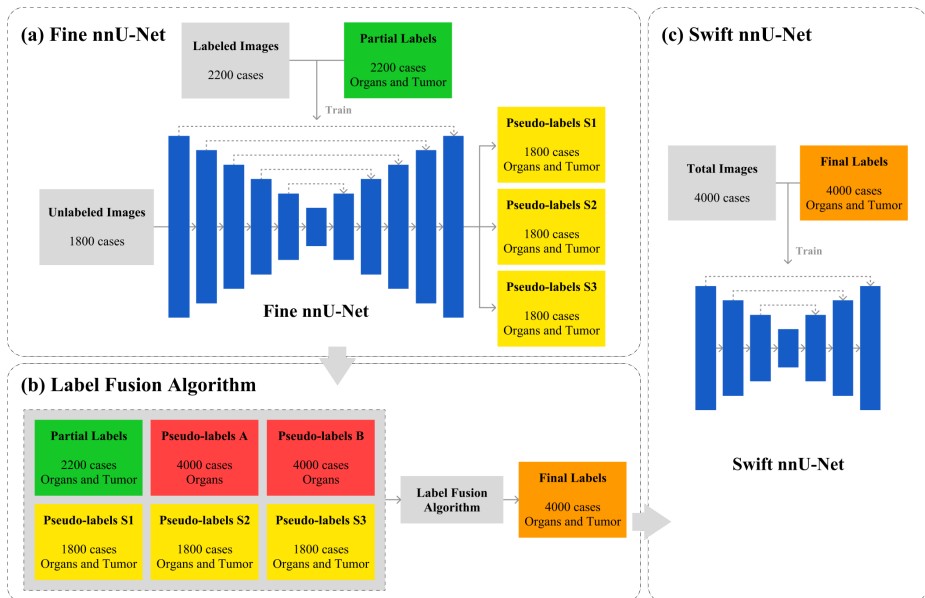

**Fig. 1.** Overall framework of the proposed method (a) Pseudo-label unlabeled images with Fine nnU-Net, which is designed to perform fine segmentation, and the pseudo-labels created by our Team Snuhmedisc are called Pseudo-labels S1, S2, and S3, respectively. (b) Create Final Labels to be used in the final model based on the algorithm designed by our team using the provided or generated labels. (c) Train Swift nnU-Net, a final model designed to make efficient inference based on the Final Labels created for all images, including unlabeled images.

## 2   Method

In this study, we designed two 3D nnU-Nets for effective training and inference. Our framework consists of three steps, as depicted in Fig. 1. (a) using the Fine nnU-Net to perform pseudo-labeling on unlabeled images; (b) applying the Label Fusion Algorithm to build the training dataset of the final model; and (c) training the Swift nnU-Net based on the final dataset and performing an efficient inference. Each nnU-Net model has adjustable hyperparameters to improve its accuracy and efficiency.

We used three labels provided by FLARE23. (1) Partial Labels, which were partially labeled out of 14 classes consisting of 13 organs and 1 tumor; (2) Pseudo-labels A, based on the model of Team Aladdin5 [22], which had the highest dice similarity coefficient (DSC) in FLARE22; and (3) Pseudo-labels B, based on the model of Team Blackbean [14], which had the highest normalized surface distance (NSD). Example images and descriptions of each label are shown in Fig. 2.

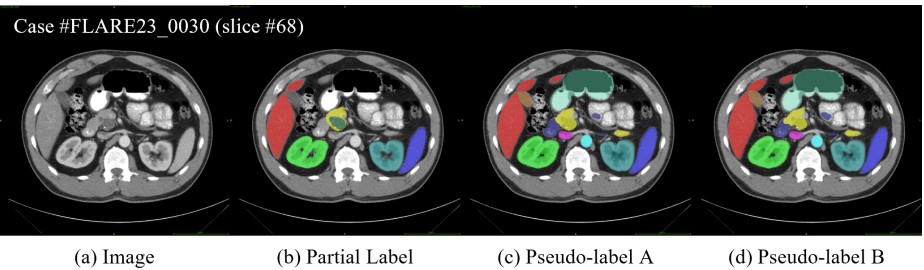

(a) Image          (b) Partial Label          (c) Pseudo-label A          (d) Pseudo-label B

**Fig. 2.** Examples of the three types of labels provided for the training dataset: (a) Slice #68 image from Case #FLARE23_0030, (b) Partial Labels, which are only partially annotated for 14 classes, (c) Pseudo-labels A for 13 organs generated from Team Aladdin5's model with the best DSC on FLARE22, and (d) Pseudo-labels B for 13 organs generated from Team Blackbean's model with the best NSD on FLARE22.

### 2.1   Preprocessing

Preprocessing was performed using similar techniques as those for nnU-Net. The preprocessing cropped the image to include crucial regions or regions of interest, resampling to ensure that all image pixels were equally spaced according to the target spacing, and normalization to ensure consistency in the intensity range of pixels in the image. On the other hand, we didn't conduct any postprocessing in our settings.

## 2.2   Proposed Method

**Fine nnU-Net**  The Fine nnU-Net is a model designed to generate high-quality pseudo-labels S1, S2, and S3 for unlabeled images and is trained with Partial Labels. Precise segmentation is crucial because the generated pseudo-labels are used as training data for the final model through the Label Fusion Algorithm. The hyperparameters of nnU-Net was tuned to effectively perform the segmentation of abdominal organs and tumors in abdominal CT scans. The values used in the Fine nnU-Net are listed in Table 1.

**Table 1.** Fine nnU-Net Hyperparameters

| | |
|---|---|
| Base number of features | 32 |
| Patch size | [56, 224, 224] |
| Target spacing | [2.50, 0.80, 0.80] |
| Number of stages | 6 |
| Convolution kernel sizes | [[3,3,3], [3,3,3], [3,3,3], [3,3,3], [3,3,3], [3,3,3]] |
| Pooling operation kernel sizes | [[1,2,2], [2,2,2], [2,2,2], [2,2,2], [1,2,2]] |

**Label Fusion Algorithm**  The Label Fusion Algorithm presented in this study was designed to collect the benefits of the provided datasets to form a complete dataset suitable for final training. The label data used were partial labels, which contain the tumor class, compared to other labels, such as pseudo-labels A and B, which ensure high performance for 13 organs, and pseudo-labels S1, S2, and S3, which were generated by our team from the three latest models of Fine nnU-Net. They are the models saved after 1000, 950, and 900 epochs. All algorithmic processing was centered on the nonzero mask regions of the provided pseudo-labels, allowing precise analysis of the abdominal region.

The Label Fusion Algorithm is designed as a different algorithm for each provided label of labeled and unlabeled images, which is divided into two parts as follows; the detailed flowchart is shown in Fig. 3.

(a)  Algorithm for Labeled Images

- Organs (class 1-13)

  1. Pseudo-labels A and B was used because of their proven performance for organs. The organ labels are fused using a union operation, and the union operation is chosen because both pseudo-labels are generated from the models with the best DSC and NSD.
  2. If pseudo-labels A and B mark different organs in a particular pixel, majority voting is performed by referring to the corresponding pixel values of the partial labels.

3. If all three labels (pseudo-labels A and B, and partial labels) point to different organs, the pixel is assigned to the background class.

- Tumors (class 14)

  1. Overlay the tumor labels with partial labels over the organ labels generated in the previous step.

(b) Algorithm for Unlabeled Images

- Organs (class 1-13)

  1. Perform a union operation on classes 1-13 of pseudo-labels A and B in the same manner as in labeled image processing.
  2. If pseudo-labels A and B represent different organs in a particular pixel, majority voting is performed by referring to the corresponding pixel value in pseudo-label S1.
  3. If all three labels (pseudo-labels A, B, and S1) pointed to different organs, the pixel was assigned to the background class.

- Tumors (class 14)

  1. Perform majority voting based on the tumor labels in pseudo-labels S1, S2, and S3.
  2. Overlay the resulting tumor labels over the organ labels generated in the previous step.

**Swift nnU-Net**  In this study, the Swift nnU-Net was proposed to achieve fast inference speed and efficient computation in a low-resource environment by modifying the existing nnU-Net structure and hyperparameters. Training was performed using the final labels for the entire image generated by the Label Fusion Algorithm. The detailed values of the model are listed in Table 2.

**Table 2.** Swift nnU-Net Hyperparameters

| | |
|---|---|
| Base number of features | 24 |
| Patch size | [32, 128, 192] |
| Target spacing | [3.00, 1.50, 1.50] |
| Number of stages | 4 |
| Convolution kernel sizes | [[3,3,3], [3,3,3], [3,3,3], [3,3,3]] |
| Pooling operation kernel sizes | [[1,2,2], [2,2,2], [2,2,2]] |

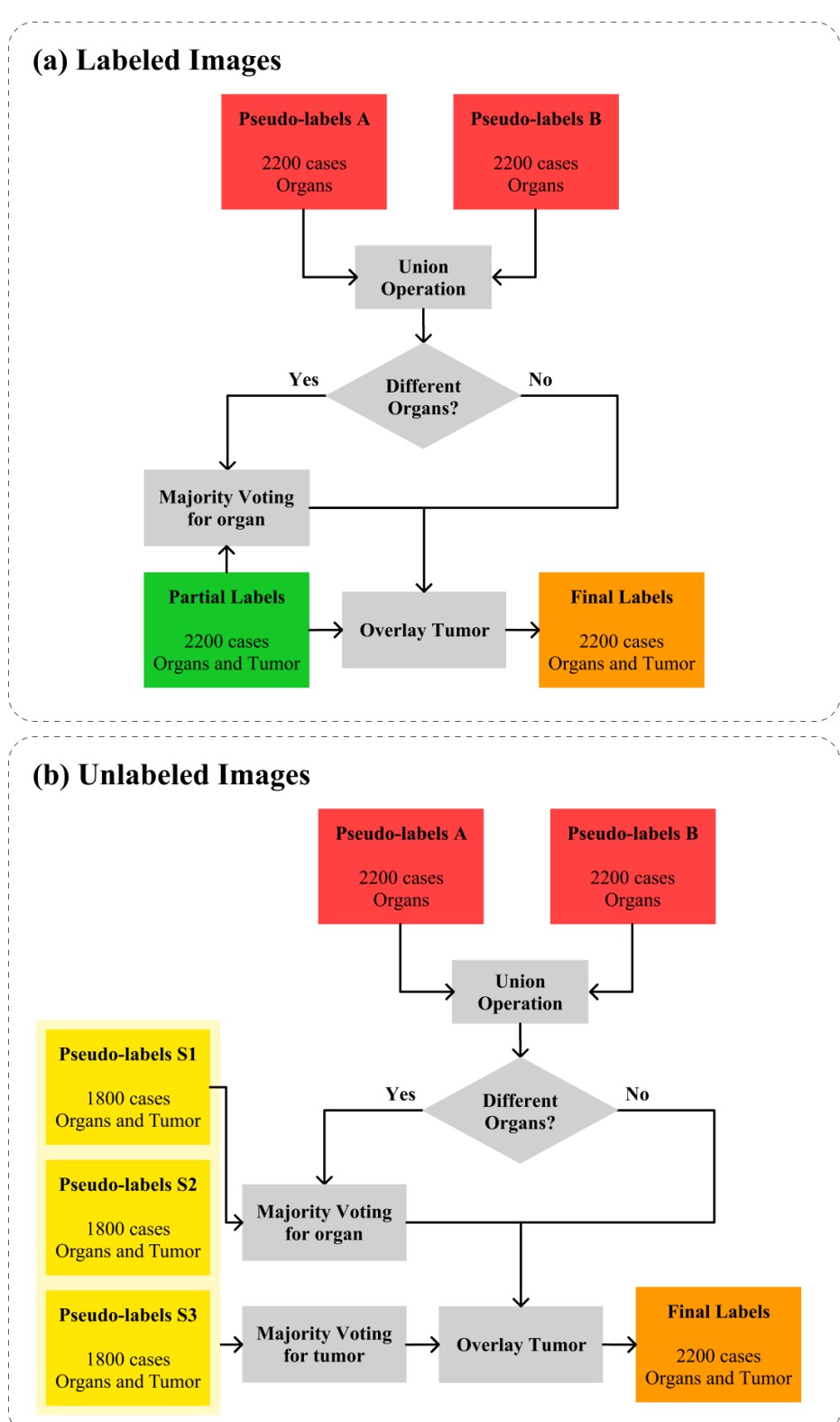

**Fig. 3.** Overall flowchart of Label Fusion Algorithm. (a) For labeled images, majority voting for organs is performed; (b) for unlabeled images, majority voting for organs and tumor is performed.

## 3    Experiments

### 3.1    Dataset and evaluation measures

The FLARE23 dataset including TCIA [4], LiTS [1], MSD [21], KiTS [11,13], autoPET [7,6], TotalSegmentator [23], and AbdomenCT-1K [18] comprises multiracial, multicenter, multidisease, multiphase, and multivendor CT images collected from over 30 medical centers under the license permission. The training dataset consisted of 4,000 images, of which 2,200 were labeled images with partial labels, and the remaining 1,800 were unlabeled images.

The partial labels consisted of 14 classes, 13 organs and 1 tumor, but only limited targets were annotated. This partial labeling setup is consistent with real-world applications because many medical institutions only focus on specific organs or tumors. The annotation process used ITK-SNAP [26], nnU-Net [15], and MedSAM [17].

In addition, two types of pseudo-labels were provided, generated based on models that performed well in FLARE22 on all 4000 training images and consisting of classes for 13 organs. Because last year's challenge was to segment only 13 organs without tumor class, the validation dataset consisted of 100 images and 400 test datasets.

These evaluation measures can be classified as accuracy and efficiency. The evaluation metrics related to accuracy are the DSC, which shows the overlap between the ground truth and the prediction, and the NSD, which shows the similarity between the outer boundaries of the ground truth and the prediction. The DSC and NSD for the 13 organ classes and the DSC and NSD for the tumor class were separated and used as evaluation metrics.

The efficiency-related evaluation metrics included the running time and area under the GPU memory-time curve. The running time was 15 s for each case, and the GPU consumption reached 4 giga byte (GB).

### 3.2    Implementation details

**Data augmentation**  We used augmentation techniques such as elastic deformation, rotation, scaling, brightness and contrast adjustment, and gamma transformation during the training process. Moreover, we applied test time augmentation (TTA) only for inference in the Fine nnU-Net.

**Environment settings**  The development environment and requirements are presented in Table 3.

**Table 3.** Development environments and requirements

| | |
|---|---|
| System | Ubuntu 18.04.6 LTS |
| CPU | AMD EPYC 7402 2P 24-Core Processor CPU@2.8GHz |
| RAM | 64×8GB; 3200MT/s |
| GPU (number and type) | One NVIDIA RTX A6000 D6 48GB |
| CUDA version | 11.3 |
| Programming language | Python 3.7.13 |
| Deep learning framework | torch 1.12.0, torchvision 0.13.0 |
| Specific dependencies | nnU-Net 1.7.0 |
| Code | |

**Training protocols** The training protocols of Fine nnU-Net and Swift nnU-Net are listed in Table 4. and 5. respectively.

**Table 4.** Training protocols for Fine nnU-Net

| | |
|---|---|
| Network initialization | "He" normal initialization |
| Batch size | 2 |
| Patch size | 56×224×224 |
| Total epochs | 1000 |
| Optimizer | SGD with nesterov momentum ($\mu = 0.99$) |
| Initial learning rate (lr) | 0.01 |
| Lr decay schedule | reduced by 10% every 200 epochs |
| Training time | 40 hours |
| Number of model parameters | 87.22M |
| Number of flops | 497T |
| $CO_2$eq | 5.7kg |

**Table 5.** Training protocols for Swift nnU-Net

| | |
|---|---|
| Network initialization | "He" normal initialization |
| Batch size | 2 |
| Patch size | 32×128×192 |
| Total epochs | 1500 |
| Optimizer | SGD with nesterov momentum ($\mu = 0.99$) |
| Initial learning rate (lr) | 0.01 |
| Lr decay schedule | reduced by 17.81% every 200 epochs |
| Training time | 15.43 hours |
| Number of model parameters | 3.15M |
| Number of flops | 62.17G |
| $CO_2$eq | 2.20kg |

## 4    Results and Discussion

### 4.1    Quantitative results on validation set

In this study, an experiment was conducted to verify the effectiveness of using unlabeled images with the proposed Label Fusion Algorithm. Based on our baseline model, we compared the training results with the Label Fusion Algorithm on 2200 labeled images with those on all 4000 images, including unlabeled images. Furthermore, when using the entire image, the performance improved by 0.56% in DSC and 0.22% in NSD for organs, and 9.85% in DSC and 6.57% in NSD for tumors. The results are presented in Table 6.

**Table 6.** Unlabeled images ablation study

| Target | Labeled Only | | With Unlabeled | |
|---|---|---|---|---|
| | DSC(%) | NSD(%) | DSC(%) | NSD(%) |
| Liver | 97.20 | 98.33 | 97.27 | 98.70 |
| Right Kidney | 93.11 | 94.67 | 93.14 | 93.65 |
| Spleen | 96.49 | 98.11 | 95.71 | 97.66 |
| Pancreas | 83.85 | 95.57 | 84.17 | 95.84 |
| Aorta | 95.34 | 98.35 | 95.36 | 98.81 |
| Inferior vena cava | 90.43 | 92.33 | 91.66 | 93.71 |
| Right adrenal gland | 82.48 | 95.39 | 83.74 | 95.80 |
| Left adrenal gland | 81.73 | 94.08 | 83.78 | 95.15 |
| Gallbladder | 83.10 | 83.59 | 82.83 | 83.48 |
| Esophagus | 80.96 | 92.10 | 81.64 | 93.00 |
| Stomach | 92.66 | 96.42 | 93.06 | 96.31 |
| Duodenum | 80.83 | 94.29 | 82.86 | 95.32 |
| Left kidney | 92.20 | 93.83 | 92.46 | 92.47 |
| Tumor | 26.15 | 19.46 | 36.00 | 26.03 |
| Organs Average | 88.49 | 94.39 | 89.05 | 94.61 |
| Total Average | 84.04 | 89.04 | 85.26 | 89.71 |

After verifying the effectiveness of using unlabeled images, we conducted experiments on hyperparameters that could balance the efficiency and accuracy for all 4000 images. The experimental results, based on the determined hyperparameters, are listed in Table 7. and Table 8.

**Table 7.** Final DSC, NSD results for validation set

| Target | Public Validation | | Online Validation | |
|---|---|---|---|---|
| | DSC(%) | NSD(%) | DSC(%) | NSD(%) |
| Liver | $97.42 \pm 0.68$ | $98.63 \pm 1.67$ | 97.23 | 98.28 |
| Right Kidney | $94.16 \pm 6.89$ | $93.81 \pm 8.89$ | 93.39 | 92.82 |
| Spleen | $96.37 \pm 2.21$ | $97.42 \pm 5.09$ | 95.99 | 97.37 |
| Pancreas | $85.74 \pm 5.42$ | $96.84 \pm 4.77$ | 84.26 | 95.82 |
| Aorta | $94.96 \pm 3.05$ | $98.63 \pm 3.21$ | 95.31 | 98.95 |
| Inferior vena cava | $91.84 \pm 4.06$ | $94.39 \pm 4.58$ | 91.55 | 93.98 |
| Right adrenal gland | $88.05 \pm 4.04$ | $98.60 \pm 1.54$ | 87.20 | 98.25 |
| Left adrenal gland | $87.94 \pm 3.45$ | $98.00 \pm 2.33$ | 86.65 | 96.69 |
| Gallbladder | $83.12 \pm 24.77$ | $84.58 \pm 26.01$ | 82.02 | 83.18 |
| Esophagus | $80.76 \pm 16.32$ | $91.47 \pm 15.71$ | 81.54 | 92.50 |
| Stomach | $92.83 \pm 4.78$ | $95.32 \pm 6.71$ | 93.23 | 95.92 |
| Duodenum | $82.31 \pm 7.41$ | $94.65 \pm 5.33$ | 82.90 | 94.93 |
| Left kidney | $92.45 \pm 9.56$ | $91.48 \pm 11.93$ | 92.96 | 92.06 |
| Tumor | $38.38 \pm 32.01$ | $27.61 \pm 25.53$ | 35.70 | 25.52 |
| Organs Average | $89.84 \pm 8.70$ | $94.90 \pm 7.60$ | 89.56 | 94.67 |
| Total Average | $86.17 \pm 17.65$ | $90.10 \pm 20.61$ | 85.71 | 89.73 |

**Table 8.** Quantitative evaluation of segmentation efficiency in terms of the running time and GPU memory consumption. Total GPU denotes the area under GPU Memory-Time curve. Evaluation GPU platform: NVIDIA QUADRO RTX5000 (16G)

| Case ID | Image Size | Running Time (s) | Max GPU (MB) | Total GPU (MB) |
|---|---|---|---|---|
| 0001 | (512, 512, 55) | 27.41 | 2626 | 20535 |
| 0051 | (512, 512, 100) | 18.13 | 2060 | 21754 |
| 0017 | (512, 512, 150) | 20.39 | 2060 | 21826 |
| 0019 | (512, 512, 215) | 32.95 | 2060 | 23259 |
| 0099 | (512, 512, 334) | 20.82 | 2060 | 22904 |
| 0063 | (512, 512, 448) | 24.9 | 2060 | 27210 |
| 0048 | (512, 512, 499) | 26.48 | 2060 | 29680 |
| 0029 | (512, 512, 554) | 30.57 | 2060 | 32723 |

## 4.2   Qualitative results on validation set

Examples of the segmentation results based on the Swift nnU-Net are shown in Fig. 4. These are the final segmentation results after training with 4000 datasets using the Label Fusion Algorithm and applying strategies for fast inference. In Cases 7 and 77, all large and small organs and tumors were well segmented. However, for Case 13, insignificant tumors were not well predicted, and for Case 67, predicting tumors in organs that do not belong to the 13 organs being segmented is challenging.

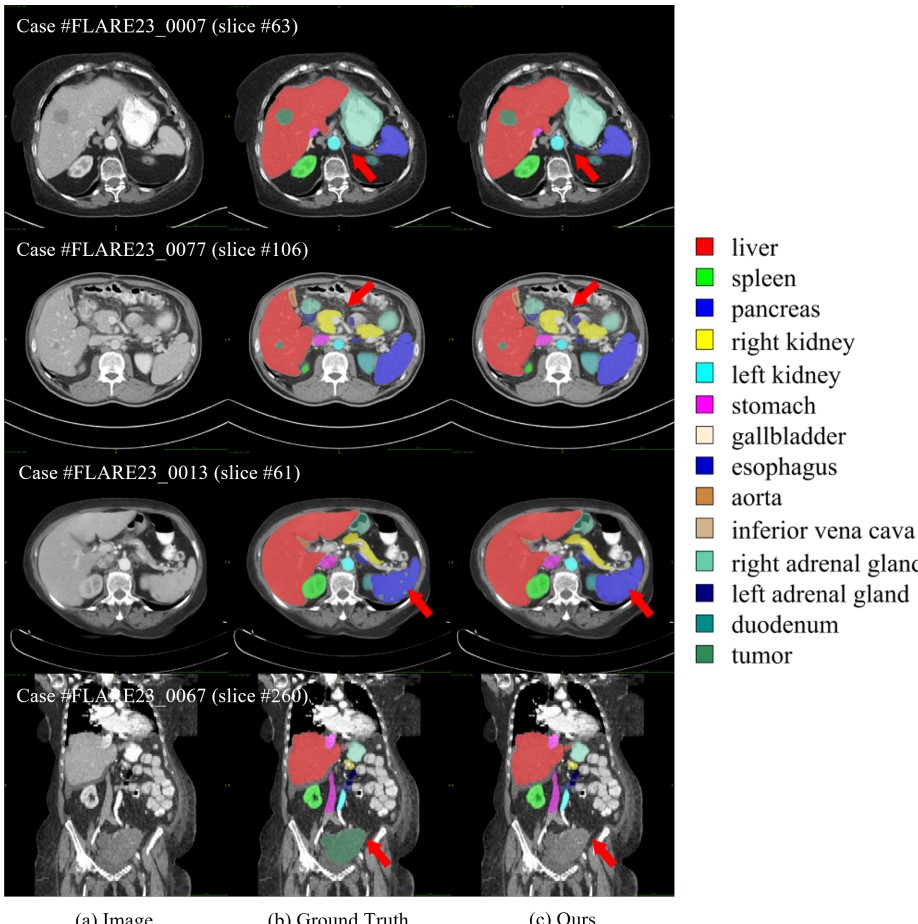

(a) Image          (b) Ground Truth          (c) Ours

**Fig. 4.** Qualitative results of our Swift nnU-Net. Examples of good segmentation of organs and tumors were Case #FLARE23_0007 and Case #FLARE23_0077, whereas challenging cases were Case #FLARE23_0013 and Case #FLARE23_0067.

### 4.3   Segmentation efficiency results on validation set

Efficiency experiments on the final submitted Docker were conducted using a GPU: One NVIDIA GeForce RTX 3070 8G, CPU: AMD Ryzen 7 5800X 8-Core Processor CPU@3.80GHz, RAM:16×4GB; 3200MT/s. Segmentation efficiency was measured using the official evaluation code for 100 validation cases, and the results are listed in Table 9.

**Table 9.** Efficiency evaluation results of our submitted docker. All metrics reported are the average values on 100 validation cases

| Time | Max GPU Memory | AUC GPU Time |
|------|----------------|--------------|
| 10.7s | 3344.9MB | 20316.72MB |

### 4.4   Results on final testing set

Based on our methodology, experiments were conducted on 400 final testing sets. In terms of efficiency, the inference time was an average of 14.03 seconds and the AUC GPU Time was 15400MB. In terms of accuracy, the DSC values for organs and tumors segmentation were 89.83% and 37.36%, respectively, and the NSD values were 95.00% and 24.53%. Table 10 and Table 11 show the detailed results.

**Table 10.** Final DSC, NSD results for testing set

| Target | Testing | |
|--------|---------|---------|
|  | DSC(%) | NSD (%) |
| Liver | 96.27 | 96.95 |
| Right Kidney | 93.94 | 93.09 |
| Spleen | 96.05 | 97.61 |
| Pancreas | 87.93 | 96.83 |
| Aorta | 95.37 | 99.52 |
| Inferior vena cava | 92.10 | 95.14 |
| Right adrenal gland | 83.41 | 96.18 |
| Left adrenal gland | 84.06 | 95.60 |
| Gallbladder | 80.91 | 83.76 |
| Esophagus | 86.32 | 96.46 |
| Stomach | 93.19 | 95.84 |
| Duodenum | 85.38 | 95.84 |
| Left kidney | 92.81 | 92.15 |
| Tumor | 37.36 | 24.53 |
| Organs Average | 89.83 | 95.00 |
| Total Average | 86.08 | 89.96 |

**Table 11.** Final efficiency for testing set

| Time | AUC GPU Time |
|------|--------------|
| 14.03s | 15400MB |

### 4.5   Limitation and future work

The main limitation of this study was the low performance of tumor segmentation compared to that of organs. Because the features of tumors were obtained using only partially labeled data than fully labeled data, this limited the ability to achieve high performance for tumor segmentation. In future work, we will investigate techniques for improving the performance of tumor segmentation, particularly for abdominal organ and tumor segmentation problems in a partially labeled environment.

## 5   Conclusion

This study aimed to address the problem of abdominal organ and pan-cancer segmentation in CT images using partially labeled datasets, unlabeled images, and pseudo-labels in medical imaging, where generating fully labeled datasets is challenging. The Fine nnU-Net and Label Fusion Algorithm for the precise pseudo-labeling of unlabeled images and the Swift nnU-Net for efficient inference were proposed. Experiments for accuracy and efficiency verified the effectiveness of the proposed method, its utilization of partial labels, unlabeled images, and efficient inference strategies. Our proposed methodology with innovative framework will be a crucial step towards more precise and efficient approaches for medical imaging environment with low computational resource.

**Acknowledgements** We declare that the segmentation method we implemented for participation in the FLARE 2023 challenge has not used any pre-trained models nor additional datasets other than those provided by the organizers. The proposed solution is fully automatic without any manual intervention. We thank all the data owners for making the CT scans publicly available and CodaLab [20] for hosting the challenge platform.
This work was supported by Institute of Information & communications Technology Planning & Evaluation (IITP) grant funded by the Korea government(MSIT) (No. 2021-0-0052, Cloud-based XR content conversion and service technology development that changes according to device performance) and Institute of Information and Communications Technology Planning and Evaluation (IITP) grant funded by the Korea Government (MSIT) (No. 2021-0-00312, development of non-face-to-face patient infection activity prediction and protection management SW technology at home and community treatment centers for effective response to infectious disease).

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

**Table 12.** Checklist Table. Please fill out this checklist table in the answer column.

| Requirements | Answer |
| --- | --- |
| A meaningful title | Yes |
| The number of authors (≤6) | 5 |
| Author affiliations, Email, and ORCID | Yes |
| Corresponding author is marked | Yes |
| Validation scores are presented in the abstract | Yes |
| Introduction includes at least three parts: background, related work, and motivation | Yes |
| A pipeline/network figure is provided | Fig. 1 |
| Pre-processing | 4 |
| Strategies to use the partial label | 5 |
| Strategies to use the unlabeled images. | 5 |
| Strategies to improve model inference | 6 |
| Post-processing | 4 |
| Dataset and evaluation metric section is presented | 8 |
| Environment setting table is provided | Table 3 |
| Training protocol table is provided | Table 4, 5 |
| Ablation study | 10 |
| Efficiency evaluation results are provided | Table 8, 9 |
| Visualized segmentation example is provided | Fig. 4 |
| Limitation and future work are presented | Yes |
| Reference format is consistent. | Yes |