# OpenReview forum: "Multi-Organ and Pan-cancer Segmentation Framework from Partially Labeled Abdominal CT Datasets: Fine and Swift nnU-Nets with Label Fusion"
_MICCAI.org/2023/FLARE — Submitted to FLARE 2023_

### Official Review · Reviewer_HD4o · 2023-09-19
**Multi-Organ and Pan-cancer Segmentation Framework from Partially Labeled Abdominal CT Datasets: Fine and Swift nnU-Nets with Label Fusion**

**Rating:** 7
**Confidence:** 4

**Review:**

Strengths:
- The proposed method achieves accurate and efficient segmentation of abdominal organs and tumors. In online validation, the DSC for organs and tumors segmentation are 89.56% and 35.70%, respectively, and the NSD values are 94.67% and 25.52%, respectively. The inference time was an average of 10.7 seconds and the area under the GPU memory time curve was an average of 20316.72 MB.

Weaknesses:
- In abstract, please add “MB” after “20316.72”.
- Please enlarge the text in Fig.1.
- The descriptions of Fine nnU-Net and Swift nnU-Net are not clear. What’s the difference between Fine nnU-Net and conventional nnU-Net? What’s the structure of the Swift nnU-Net?
- Fine nnU-Net is trained with Partial Labels. How do you process organs without annotations? Is Fine nnU-Net trained using organs without annotations?
- What’s the difference among pseudo labels S1, S2, and S3? Please introduce generation process of pseudo labels S1, S2, and S3 in detail, including training data, training model, and training strategy and so on.
- In label fusion algorithm, (a) algorithm for labeled images, organs 2. , how do you define pseudo labels of organs that are without annotations in partial-label data?
- Please adjust Fig.2 to make it compact and exact.
- “after processing for organs” in annotation of Fig.2 is not descriped clear.
- Add arrows in Fig.4 to indicate regions that are well segmented or not well segmented.

---

> ### Author Response · Authors · 2023-11-15
>
> We are truly thankful for the comprehensive and enlightening review provided.
>
> - In abstract, please add “MB” after “20316.72”.
>
> : We added all "MB" for GPU memory time curve performance(p.1 Abstract). Many thanks!
>
> - Please enlarge the text in Fig.1.
>
> : We increased the font size(p.3 Fig. 1). Much appreciated!
>
> - The descriptions of Fine nnU-Net and Swift nnU-Net are not clear. What’s the difference between Fine nnU-Net and conventional nnU-Net? What’s the structure of the Swift nnU-Net?
>
> : The structure of our Fine nnU-Net and Swift nnU-Net can be seen in overall framework figure(p.3 Fig. 1), and the modified hyperparameters provided in Table 1 in page 5 and Table 2 in page 6. Thank you kindly.
>
> - Fine nnU-Net is trained with Partial Labels. How do you process organs without annotations? Is Fine nnU-Net trained using organs without annotations?
>
> : It is correct that our Fine nnU-Net was trained with only provided partial labels which were partially labeled out of 14 classes consisting of 13 organs and 1 tumor. Thanks a ton!
>
> - What’s the difference among pseudo labels S1, S2, and S3? Please introduce generation process of pseudo labels S1, S2, and S3 in detail, including training data, training model, and training strategy and so on.
>
> : Pseudo-labels S1, S2, and S3 are the pseudo-labels generated by the three latest models saved during the training process of Fine nnU-Net. They are the models saved after 1000, 950, and 900 epochs, respectively, and Fine nnU-Net was trained with the provided partial labels. Heartfelt thanks.
>
> - In label fusion algorithm, (a) algorithm for labeled images, organs 2. , how do you define pseudo labels of organs that are without annotations in partial-label data?
>
> : The pseudo-labels A and B used by the algorithm for labeled images to process organs are pseudo-labels generated by models from FLARE22's Team aladdin5 and blackbean, respectively. Our sincere thanks.
>
> - Please adjust Fig.3 to make it compact and exact.
>
> : We revised the figure to make it more concise and clear(p.7 Fig. 3). We're truly grateful for your support.
>
> - “after processing for organs” in annotation of Fig.3 is not descriped clear.
>
> : We removed the phrase "after processing for organs" as we felt it could be confusing(p.7 Fig. 3). Grateful for your help.
>
> - Add arrows in Fig.4 to indicate regions that are well segmented or not well segmented.
>
> : We've added arrows pointing to the regions that are well segmented and the parts that are not(p.12 Fig.4). Deeply thankful.

---

### Official Review · Reviewer_purc · 2023-09-21
**Multi-Organ and Pan-cancer Segmentation Framework from Partially Labeled Abdominal CT Datasets: Fine and Swift nnU-Nets with Label Fusion**

**Rating:** 7
**Confidence:** 5

**Review:**

Summary:

The author has proposed a label fusion strategy for partially labeled and unlabeled data, and ultimately employed a distillation approach to learn from the pseudo-labels, achieving impressive segmentation results.

Comments:

1、The author's description of the generation methods and their distinctions for pseudo-labels is rather limited and would benefit from additional elaboration. It is important to provide further details in this regard.

---

> ### Author Response · Authors · 2023-11-15
>
> We are sincerely grateful for the insightful feedback provided by the reviewer. We have revised the illustration of the specific algorithm for generating pseudo-labels to make it easier and more sophisticated(p.7 Fig. 3), and added a description of the generation method(p.5).

---

### Official Review · Reviewer_vF7u · 2023-09-26
**good paper overall**

**Rating:** 7
**Confidence:** 4

**Review:**

Pros: The paper proposes a method utilizing Fine nnU-Net for pseudo-labeling unlabeled images, a Label Fusion Algorithm to combine different labels, and Swift nnU-Net for efficient inference. The paper is well-structured and flows smoothly, making it easy to follow the author's arguments.

Cons: In the Qualitative analysis section, fig. 4, it is not clear what the correspondence between classes and represented colors. Please label them.

---

> ### Author Response · Authors · 2023-11-15
>
> Our deepest thanks go to the reviewer for their constructive and valuable comments. Based on your comments, we have added a legend to the labels indicating which class each color represents in the Qualitative results section(p.12 Fig. 4).

---

### Official Review · Reviewer_M9EG · 2023-10-03
**Multi-Organ and Pan-cancer Segmentation Framework from Partially Labeled Abdominal CT Datasets: Fine and Swift nnU-Nets with Label Fusion**

**Rating:** 7
**Confidence:** 5

**Review:**

good accept

---

> ### Author Response · Authors · 2023-11-15
>
> Thank you for your comments.

---

### Official Review · Reviewer_9StH · 2023-10-04
**This study uses a method combining nnUNet and Swin nnUNet to segment organs and tumours and achieves an average DSC score of 89.56% for organs and 35.70% for tumours on the online validation leaderboard.**

**Rating:** 7
**Confidence:** 5

**Review:**

This paper is well organized and shows enough details of the method. One of the main contributions of the proposed method is to propose a label fusion strategy to improve the segmentation accuracy. However, the accuracy of the proposed method is low, especially in tumour segmentation tasks. In addition, the font in the picture is too small.

---

> ### Author Response · Authors · 2023-11-15
>
> We appreciate the reviewer’s comment. We increased the font size of the picture(p.3 Fig. 1, p.7 Fig. 3).

---

> > ### Comment · Reviewer_9StH · 2023-11-30
> >
> > The revised paper expresses more clearly, which is conducive to reading. It looks good overall and contains enough experimental detail.

---

### Decision · Program_Chairs · 2023-10-24

Accept